# Chemical Profile of Essential Oils of Selected Lamiaceae Plants and In Vitro Activity for Varroosis Control in Honeybees *(Apis mellifera)*

**DOI:** 10.3390/vetsci10120701

**Published:** 2023-12-13

**Authors:** Roberto Bava, Fabio Castagna, Carmine Lupia, Stefano Ruga, Vincenzo Musella, Filomena Conforti, Mariangela Marrelli, Maria Pia Argentieri, Domenico Britti, Giancarlo Statti, Ernesto Palma

**Affiliations:** 1Department of Health Sciences, University of Catanzaro Magna Græcia, 88100 Catanzaro, Italy; roberto.bava@unicz.it (R.B.); studiolupiacarmine@libero.it (C.L.); stefano.ruga@studenti.unicz.it (S.R.); musella@unicz.it (V.M.); britti@unicz.it (D.B.); palma@unicz.it (E.P.); 2Mediterranean Ethnobotanical Conservatory, 88054 Catanzaro, Italy; 3Department of Pharmacy, Health and Nutritional Sciences, University of Calabria, 87036 Cosenza, Italy; filomena.conforti@unical.it (F.C.); mariangela.marrelli@unical.it (M.M.); g.statti@unical.it (G.S.); 4Department of Pharmacy-Drug Sciences, University of Bari Aldo Moro, 70125 Bari, Italy; mariapia.argentieri@uniba.it; 5Department of Health Sciences, Institute of Research for Food Safety & Health (IRC-FISH), University of Catanzaro Magna Græcia, 88100 Catanzaro, Italy; 6Nutramed S.c.a.r.l., Complesso Ninì Barbieri, 88021 Catanzaro, Italy

**Keywords:** honeybee (*Apis mellifera*), *Varroa destructor*, essential oils (EOs), *Origanum vulgare* subsp. *viridulum*, *Thymus capitatus*, *Thymus longicaulis*, *Salvia rosmarinus*, acaricidal activity

## Abstract

**Simple Summary:**

*Varroa destructor* acariasis is currently the main threat to the health and survival of honeybee colonies. Chemicals are often used to control this parasitosis. However, overuse and misuse over the years has allowed the mite to acquire resistance to synthetic active ingredients. In this scenario, it is vital to search for alternative therapeutic solutions. Essential oils (EOs) are a promising therapeutic choice, as they have a complex chemical composition, making them unlikely to be prone to the development of resistance. In addition, they are easily degraded in the environment and have a low toxicity for humans, characteristics that make them particularly attractive. In this research study, four EOs from the Lamiaceae family, isolated from botanical species native to the Calabria region, Southern Italy, were tested in contact toxicity tests against *V. destructor*. *Origanum vulgare* subsp. *viridulum*, *Thymus capitatus* and *Thymus longicaulis,* used at 2 mg/mL, were found to have a high level of efficacy, neutralizing (dead + inactivated) 94%, 92% and 94% of parasites, respectively. These EOs could be chosen and tested in subsequent in vivo studies.

**Abstract:**

The most significant ectoparasitic mite of honeybees, *Varroa destructor*, has a detrimental effect on bee health and honey output. The principal strategy used by the control programs is the application of synthetic acaricides. All of this has resulted in drug resistance, which is now a major worry for beekeeping. As a result, research on alternate products and techniques for mite management is now required. The aim of this study was to determine whether essential oils (EOs) extracted from botanical species of Lamiacae, typical of the Calabria region of Southern Italy, could reduce the population of the mite *V. destructor*. Among the best-known genera of the Lamiaceae family are oregano, rosemary and thyme, whose EOs were employed in this study. By steam distillation, the EOs were extracted from *Origanum vulgare* subsp. *viridulum* (Martrin-Donos) Nyman, *Thymus capitatus* Hoffmanns. and Link, *Thymus longicaulis* C.Presl and *Salvia rosmarinus* Schleid. plant species harvested directly on the Calabrian territory in their balsamic time. Each EO went to the test in vitro (contact toxicity) against *V. destructor*. Fifty adult female mites, five for each EO and the positive and negative control, were used in each experimental replicate. The positive controls comprised five individuals treated to Amitraz dilute in acetone, and the negative controls included five individuals exposed to acetone alone. To create the working solution to be tested (50 μL/tube), the EOs were diluted (0.5 mg/mL, 1 mg/mL, 2 mg/mL and 4 mg/mL) in HPLC-grade acetone. After 1 h of exposure, mite mortality was manually assessed. *Origanum vulgare* subsp. *viridulum*, *Thymus capitatus* and *Thymus longicaulis* were the EOs with the highest levels of efficiency at 2 mg/mL, neutralizing (dead + inactivated), 94%, 92% and 94% of parasites, respectively. *Salvia rosmarinus* EO gave a lower efficacy, resulting in a percentage of 38%. Interestingly, no adverse effects were highlighted in toxicity tests on honeybees. These results show that these OEs of the Lamiaceae family have antiparasitic action on *V. destructor*. Therefore, they could be used, individually or combined, to exploit the synergistic effect for a more sustainable control of this parasite mite in honeybee farms.

## 1. Introduction

*Apis mellifera*, also known as the Western honeybee, is one of the most significant insect species in the world [1]. In fact, its breeding provides products with important pharmaceutical and nutraceutical characteristics, such as honey, propolis, pollen and wax [2,3]. It is also prized for the pollination services [4]. Given the significance of honeybees to agriculture and ecosystems, problems that affect their health are a matter of universal concern. When compared to long-term historical averages, the annual loss of managed honeybee colonies has increased dramatically in recent years due to several reasons [5]. *Varroa destructor* is consensually recognized as the most serious cause of colony loss [6]. With its parasitization, the mite alters cuticle properties, the immunological response system and the weight and fitness of freshly emerging adult honeybees by feeding on the fat body [7,8]. Additionally, *V. destructor* is the vector of many viruses [9]. Individuals that are infected become weaker and have shorter lives, and the illness might result in the final colony collapse (CCD) [10,11,12]. The cycle of *V. destructor* is strongly connected to and timed with honeybee brood development [13]. Adult females of the mite enter the honeybee colony, carried by workers and drones, typically hiding behind the honeybees’ abdominal sternites, and enter the brood cells only a few hours before closing [14,15]. *V. destructor* has an exponential growth in the colony [16]. If infestation is not controlled, honeybee populations in colonies with heavy infestations decline dramatically and finally collapse [17]. Controlling infestation is even more important due to climate change, which leads to nutritionally impoverished ecosystems [18,19]. Several studies have shown that a diet deficient in pollen, both in quantity and quality, can interfere with the honeybees’ immune response to pathogenic noxa [20,21]. To keep mite numbers in check in temperate locations, colonies may need to be treated multiple times a year with acaricides [22]. The use of synthetic acaricides has long been seen to be the most efficient method of controlling *V. destructor*. Aside from their diminishing effectiveness brought on by the emergence of pharmacological resistance, the misuse of these active ingredients has frequently resulted in contamination of hive products, with possible implications for human and honeybee health [23,24,25,26]. The need to develop new and risk-free approaches to parasite management is growing as a result. Natural products (NPs) and their derivatives present a more appealing alternative to synthetic drugs [27,28,29,30,31]. NPs have been shown to have important pharmacological properties that have been validated in experimental studies in human and veterinary medicine [30,32,33,34,35]. Compounds belonging to the broad category of NPs are also attractive because they are often inexpensive and pose fewer health problems for both humans and honeybees [36,37]. As a result, beekeepers are increasingly interested in their use [38]. Because of this, the usage of organic acids and EOs is growing. However, a number of studies indicate that using organic acids to counteract *V. destructor* infestation may be detrimental to honeybee health [39]. For instance, an open and closed brood is observed to be damaged and removed [40,41]. In addition, irreversible damage to honeybees’ digestive and excretory organs and glands, damage to the queen or often even her premature death and a drop in the pH of honey in the following season have been reported [42,43,44]. With this in mind, EOs seem to be a better option for lowering pest populations. EOs are defined as odorous compounds that are typically produced from botanically specified plant raw materials [45]. They are typically separated from the aqueous phase by using a physical technique that has no impact on their chemical makeup [46]. Terpenoids and phenylpropanoids, monoterpenes, sesquiterpenes, aldehydes and alcohols, as well as other components, can be found in EOs [45]. EOs showed a wide range of activity, from fatal to sublethal effects against a variety of insects and mites [47,48,49,50]. EOs from a variety of plant families, including Apiaceae, Asteraceae, Chenopodiaceae, Cupressaceae, Lamiaceae, Lauraceae, Myrtaceae, Zingiberaceae, Umbelliferae and Geraniaceae, have been shown to possess acaricidal properties [49,51,52,53,54]. In fact, there has been a steady increase in recent years in the amount of scientific attention paid to the use of EOs in parasite control and pest management tactics [30,31,37,39,55,56,57]. Given that they are mixtures of many components, they have been seen to have a variety of mechanisms of action, ranging from acute toxicity to repellency, antinutritional and developmental inhibitory effects, and they influence neurological and metabolic processes [45]. As a result, resistance to these botanical derivatives has only little changed, and it can be said that these plant-derived substances are safe and effective substitutes for harmful synthetic pesticides. Therefore, EOs are among the most promising natural alternatives to chemicals since they have few negative side effects. In addition to their acaricidal actions, EOs frequently have antibacterial properties as well, which can benefit the health of honeybee colonies as a whole [58,59].

The aim of this work was to demonstrate the in vitro acaricidal efficacy of EOs extracted from botanical species belonging to the Lamiaceae family originating from the Calabria region.

## 2. Materials and Methods

### 2.1. Plants and EOs’ Extraction

In natural growing areas of the Calabria region (Southern Italy), at altitudes between 200 and 800 m above sea level (masl), the aerial parts of *Origanum vulgare* subsp. *viridulum, Thymus capitatus*, *Thymus longicaulis* and *Salvia rosmarinus* were gathered during their balsamic period (June/July). Voucher specimens of each tested species were deposited in positions 5 (*Origanum vulgare* subsp. *viridulum),* 12 *(Thymus capitatus)*, 54 (*Thymus longicaulis)* and 42 (*Salvia rosmarinus)* of the Lamiaceae family at the Mediterranean Ethnobotanical Conservatory, Sersale (CZ), Italy.

For each botanical species, fresh plant material was cleaned and subjected to a 2-h steam distillation process, using a steel extractor apparatus (Albrigi Luigi, Verona, Italy) to extract the EOs. The obtained EOs were dried over anhydrous sodium sulfate and stored at +4 °C. 

### 2.2. Gas Chromatography–Mass Spectrometry (GC-MS) Analyses

A Trace GC–FID ultra gas chromatograph (Thermo Finnigan, Bremen, Germany) was used for the chemical analysis. Each distilled EO was solubilized in hexane before analysis, and then 1 μL of EO was injected. A silica capillary column (30 m × 0.25 mm; 0.25 μm film thickness) fused with DB-5 (J&W Scientific) was employed for the cold on-column injection. Conditions for the chromatograph were as follows: 300 °C was the detector temperature, and a 4 °C min^−1^ program was used to program the column temperature from 60 °C (5 min isothermal) to 280 °C (15 min isothermal). The carrier gas was hydrogen (35 kP; 2.0 mL/min). The 32-bit computer program Chrom-Card was used to process the data. Based on the total peak regions found in the GC-FID analysis, the composition of the EOs’ components is given as a percentage. There were no correction factors used.

A Hewlett Packard 6890-5973 mass spectrometer interfaced with an HP Chemstation (Agilent Technologies, Palo Alto, CA, USA) was used to perform the GC-MS analysis. The following were the chromatographic conditions: injector temperature of 280 °C and column oven program of 60 °C (5 min isothermal) to 270 °C (30 min isothermal) at 4 °C/min. The carrier gas (helium; 1 mL/min flow rate) was used. The capillary column used was an HP-5 MS (30 m 9 0.25 mm; 0.25 µm film thickness). The MS was operated with the following parameters: vacuum, 10-5 torr; ion source temperature, 200 °C; and electron current, 34.6 µA. Mass spectra were obtained at 1 scan/s, spanning the 40–800 amu range. The electron impact mode was used by the ion source. The splitless sampling approach was used to inject samples (1 µL). The chemical composition of the analyzed EOs was determined by comparing the GC retention times of their constituents with known authentic reference compounds (purchased by Sigma-Aldrich, Milan, Italy) in combination with Kovats Indexes (KIs) and by means of reference mass spectra from standard compounds and/or from NIST mass spectral library files [60].

### 2.3. Mite Harvesting

The tests were conducted in June/July 2023 at the Institute of Research for Food Safety and Health—IRC-FSH, in the University “Magna Græcia” of Catanzaro. Two apiaries in the province of Catanzaro in the Calabria region of Southern Italy were employed as a source of mites. Acaricide treatments had not been administered to study colonies in the six months before; consequently, they were severely naturally infected with *V. destructor*. In a nutshell, numerous frames containing drone brood were moved from the apiary to the mite collection laboratory. The procedure for mite collection is outlined below. Each brood cell in the frame was opened, depriving it of the wax layer that ensures its closure, and examined. Inspectional examination of the cell included removing the pupa inside it, observing it for the presence of any mites on its body and, finally, observing the cell walls for mites above them. When the mites were observed, they were picked up with a fine-bristled paintbrush and placed in a Petri dish. To prevent malnutrition during harvesting operations, mites were supplied with a honeybee larva (5th stage) and/or a pupa. The toxicological tests were carried out right away after the mites were collected. Mites that seemed to have just molted, to be weak or to be abnormally shaped were removed before each test because they could have responded differently in bioassays.

### 2.4. Toxicity towards V. destructor

To investigate the acute toxicity of the EOs, a topical residual bioassay was used, adopting the method from Bava et al., 2021, which slightly modified the procedure of Gashout and Guzman-Novoa, 2009 [37,61]. At least one hundred and fifty mites were gathered for each daily test in order to create the experimental replicates. Serial dilutions of the four EOs were tested. In particular, the active ingredient, namely Amitraz (Merck, 45323), and the tested EOs were diluted in acetone to concentrations of 2 mg/mL, 1 mg/mL and 0.5 mg/mL. Amitraz and essential oils are, in fact, difficult to solubilize in water, but they are soluble in organic solvents, such as acetone [62,63,64,65]. For the toxicity tests, Amitraz and acetone alone were utilized as positive and negative controls. Eppendorf tubes (2 mL) were filled with 50 µL of diluted EOs and set open in the oven to allow the acetone to evaporate. To facilitate the evaporation of the acetone and to coat the tube walls with EOs, the tubes were frequently rolled on their walls. This operation was repeated for 15–20 min. It was confirmed by Gashout and Guzmán-Novoa, 2009, that the high boiling point of the tested products, which exceeds 200 °C, makes it unlikely that a substantial portion of them would have evaporated in this time frame, whereas the boiling point of acetone is lower [61,66]. A fine paintbrush was then used to carefully insert five adult female mites into the previously prepared tube for each technical replicate, as well as for the positive and negative controls. After the mites were introduced, the tubes were sealed and put in a dark chamber for incubation at 34 °C and 65% relative humidity. The humidity and temperature values set are those that are normally present at the brood-area level in the hive nest. These values were shown in earlier research to be more favorable for the growth and reproduction of *V. destructor* [37,67]. To establish acute toxicity, after 1 h of exposure, mite mortality was recorded. Therefore, the mites in each Eppendorf tube were moved to a Petri dish and observed under a stereomicroscope. If the mites did not move when touched, they were classified as dead. Mites were classified as inactive when they moved only one or more legs but could not move from their position. The inactive state was equated with death. Inactive and dead mites were considered equally neutralized. This experimental design included ten technical replicates for each EO and its different concentrations.

### 2.5. Toxicity towards Honeybees

Using a randomly selected group of individuals, the toxicity of EOs toward adult honeybees was studied. The individuals that made up the pool came from different frames in the hive; a system that allowed for a sample of honeybees of various ages was employed. Specifically, the different frames were shaken in a container, the honeybees were sprayed with water to prevent them from flying and then mixed. Samples of these honeybees were transported to the laboratory to be processed to toxicity tests. Five experimental replicates were set up for each EO. According to Bava et al., 2021, two 50 mL Falcon tubes were filled with 1.6 mL of EO diluted in acetone [37]. One test tube was filled with the same amount of acetone as a negative control. The amount of 1.6 mL to be introduced into the Falcon tube was chosen in proportion to the volume used for the mite toxicity tests in the Eppendorf tube. As in the viability tests for *V. destructor*, the Falcon tubes were rolled on their walls many times to coat them with liquid and let the acetone in the solution evaporate. Five honeybees were put into the tubes that had previously held the EO solution. Finally, the honeybee-filled Falcon tubes were moved to an incubator (34 °C and 65% relative humidity). After one hour of exposure, according to William et al. (2013), the honeybees were put in cages (cylindrical plastic box, length = 90 mm, height = 100 mm) [68]. A fifty percent sucrose solution and water feeders were installed in these cages. The following 48 h were spent monitoring the honeybees.

### 2.6. Statistical Analysis

The program GraphPad Prism 9.0 (GraphPad Softwar, Inc., La Jolla, CA, USA) was used to perform the statistical analysis. One-way ANOVA and the Bonferroni test were used for statistical analyses, where there were multiple comparisons. Using the web program Metabo-Analyst version 5.0 (http://www.metaboanalyst.ca, accessed on 16 October 2023), a principal component analysis (PCA) was performed. The integrity of the data was examined. In place of missing values, LoDs (1/5 of each variable’s lowest positive value) were used. Log transformation was used to standardize the data, and Pareto scaling was used to pretreat them. An additional method used to graph the chemical compositions of the various EOs was clustered heat mapping.

## 3. Results

### 3.1. Chemical Composition

The steam distillation of the aerial parts of *Thymus* species allowed us to obtain yields equal to 0.4% (*T. capitatus*) and 0.1% *w*/*w* (*T. longicaulis*), while the yield of the volatile oil from *Salvia rosmarinus* was equal to 0.3%. The highest EOs yield (0.8% *w*/*w*) was obtained for *Origanum vulgare* subsp. *viridulum.* The chemical constituents of the investigated EOs were identified with gas chromatography–mass spectrometry (GC-MS). As reported in Table 1, 51 compounds were identified.

Overall, 1,8-cineole (0.83–46.92%), linalool acetate (65.27% in OV sample), thymol (0.18–31.67%) and carvacrol (0.23–54.74%) were the most abundant components, even if differently distributed among the samples. Furthermore, linalool (0.28–15.84%), borneol (2.17–11.96%), carvacrol acetate (15.22%) and β-Caryophyllene (3.87–12.58%) were detected at percentages above 10%. Other components, such as α-pinene, β-pinene, γ-terpinene, α-terpineol and thymol methyl ether, were detected at percentages above 5%.

A principal component analysis (PCA) was performed in order to have a clear overview of the distribution of the secondary metabolites in the four investigated essential oils.

The data matrix consisted of 12 samples (three determinations for each EO) and 51 variables (constituent metabolites). Figure 1 reports the biplot of the scores and loading values that were obtained by considering the first and the second principal components, which explained 77.7% of the total variance (with PC1 and PC2 explaining 44.9% and 32.9%, respectively).

The biplot clearly shows the different composition of the EOs. *Salvia rosmarinus* (SR) samples, located in the upper right half of the scores and loadings biplot, were characterized by a higher content of α- and β-pinene (compound nos. 2 and 5), camphene (3), 1,8-cineole (14), camphor (22), borneol (23) and bornyl acetate (29) compared to the other essential oils.

*Origanum vulgare* subsp. *viridulum* (OV) was instead located in the lower part of the plot, since it is mostly characterized by the presence of linalool acetate (28) and also β-ocimene (15), linalool (20), 1-octen-3-ol acetate (21), carvacrol methyl ether (27), linalool acetate (28), geranyl acetate (36) and β-bluebonnet (37).

Furthermore, the two *Thymus* species, *T. capitatus* (TC) and *T. longicaulis* (TL), were clearly discriminated in the upper left half of the plot and characterized by the highest content of carvacrol (31) and other constituents, such as *o*-cymene (12), γ-terpinene (16), *cis*-sabinene hydrate (17) and terpinen-4-ol (24).

The different distribution of the secondary metabolites is also visualized in the heatmap reported in Figure 2.

### 3.2. V. destructor Toxicity

The percentages and standard deviations (SDs) of the neutralization of the *V. destructor* parasite at the concentrations of 0.5, 1 and 2 mg/mL with each EO, acetone (negative control) and Amitraz (positive control) are presented in Figure 3 and Table 2, while Figure 2 represents the effects of the EO on neutralization of mites.

*Thymus capitatus*, *Thymus longicaulis* and *Origanum vulgare* sbps. *viridulum* concentrations of 0.5, 1 and 2 were found to be more effective acaricides than Amitraz 0.5. Furthermore, the three different concentrations achieved a mite neutralization comparable to that obtained with Amitraz 1 and 2. *Salvia rosmarinus*, on the other hand, already at a concentration of 0.5, was not comparable to the effectiveness of Amitraz 0.5.

### 3.3. Toxicity towards Honeybees

The mortality of the honeybees, exposed to the EO-treated surface, was assessed at 4, 8, 24 and 48 h. No mortality or abnormal behavior was noted during the 48 h of registration.

## 4. Discussion

Acute or chronic negative effects on mammals and non-target organisms, including birds, bees, parasitoids, and predators, as well as the emergence of pest resistance, have all been linked to the overuse of synthetic chemicals in pest management programs [69,70,71]. Residues have also been found on food and drinking-water supplies [72]. This situation has led scientists to concentrate their efforts on the search for plant-derived EOs from various plant genera and families. Plant EOs are harmless to mammals and other vertebrate animals [73,74]. Moreover, because they have a complex chemical composition and produce different mechanisms of toxic action, it is quite unlikely that parasites will develop resistance to them [45]. The Lamiaceae species were the focus of the current research since they are more accessible and widespread than other botanical species [75]. The EOs from Lamiaceae plants and their constituents, which have a variety of lethal and sublethal effects against various harmful insects and mites in the field, greenhouse and storage conditions, have great potential for use in pest management strategies and are regarded as safe, readily available and environmentally friendly alternatives to synthetic chemicals. In this research, the EOs that showed the most effectiveness were those of *Origanum vulgare* subsp. *viridulum, Thymus capitatus, Thymus longicaulis* and *Salvia rosmarinus.* When compared to the negative control (only acetone), all the EOs tested showed a high degree of effectiveness, and even at 0.5 mg/mL, their efficacy was statistically significant. Compared to the positive control, *Thymus capitatuts*, *Thymus longicaulis* and *Origanum vulgare* EOs, at different concentrations, were more effective or comparable to Amitraz. Only *Salvia rosmarinus* resulted in a lower efficacy compared to the lesser Amitraz concentration. The ability of thyme EO to control *V. destructor* infestation is well known. The thyme species most frequently studied for its acaricidal efficacy is *Thymus vulgaris* [76,77]. In this study, it was decided to focus on the species *Thymus capitatus* and *Thymus longicaulis*. In a recent study by Ghasemi et al., 2016, cage fumigation trials against *V. destructor* were conducted using the EO of *Thymus kotschyanus* [78]. The EO caused a mild mortality rate of 54.4% and 84.43% after 5 and 10 h of fumigation. In the study by Damiani et al., 2009, the efficacy of *Thymus vulgaris* was tested in a total exposure experiment. LC50, after 72 h of exposure, was 2.93 (2.27–3.53) [77]. However, the experiment most similar to the one conducted in this study is that of Hybl et al., 2021 [39]. The authors equally used a residual contact toxicity test. Also, in Hybl et al.’s 2021 experiment, thyme EO proved to be among the EOs with the best mite abatement rates [39]. The EO was able to kill 100% of the mites after 2 h of exposure. In the same study, the EO of oregano was evaluated, which also resulted in the death of 100% of the mites after two hours [39]. The EO of oregano, specifically *O. heracleoticum*, had already been assayed by our research group. In the previous experiment, the EO also proved to have an excellent ability to control *V. destructor*, neutralizing 90.9% of the mites tested at a concentration of 2 mg/mL. In our studies, the lower-efficacy results found compared to Hybl et al., 2021 [39], can be attributed to the shorter exposure times of mites to EOs. In fact, we chose to test for death after one hour and not two. This choice was dictated by the consideration that mites suffer from malnutrition and dehydration when kept more than four hours away from their natural environment, as pointed out by Milani et al., 2001 [79]. Furthermore, it is important to say that the results of research conducted with the same botanical species in different studies could lead to different results because the essential oils extracted from a botanical species have a different composition that is influenced by the characteristics of the soil in which the plant grows, the exposure to the sun, and the adaptations that the plant uses to grow better in the environment [37].

In general, our results confirm and are in accordance with those of Koc et al. (2013) [80], who reported that EOs with higher concentrations of carvacrol have greater acaricidal action. In the EOs of the species belonging to the Lamiaceae family, the most representative molecule is carvacrol, a monoterpene produced from γ-pinene [81]. According to Koc et al., 2013, EOs with higher concentrations of carvacrol have greater acaricidal action [80]. Studies have shown that low amounts of this molecule are effective in killing several mite species, thus demonstrating its toxicity [80,82,83]. Also, for these reasons, four EOs—*Origanum vulgare* subsp. *viridulum* (Martrin-Donos) Nyman, *Thymus capitatus* Hoffmanns. and Link, *Thymus longicaulis* C. Presl, and *Salvia rosmarinus* Schleid.—were chosen. The three investigated EOs containing this compound showed the best activity, while, on the contrary, lower biological effects were observed for *Salvia rosmarinus* essential oil, which lacked this constituent (Table 1). Due to the well-established connections between these molecules and plant defense, we can argue that the poisonous action of these monoterpenes in combination with other substances, such as phenylpropanoids, determined the acaricidal activity of the EOs. In a previous study, we demonstrated how the interaction between and synergistic action of the compounds that make up the phytocomplex give the EOs greater efficacy than the individual components [56]. However, when compounds with particular efficacy are found in nature, as in the case of carvacrol, one could consider isolating this molecule to exploit its high acaricidal action. Continuing our reflection, it must be said that, when designing a field formulation, one limitation of the EOs and carvacrol that should be contained is their high volatility and poor solubility in water. These characteristics make administration difficult. In order to make them more soluble and to provide a delivery shuttle that can pass across biological membranes, β-cyclodextrins (β-CD) complexation has shown to be a successful technique. Cyclodextrin complexes offer a very interesting means of overcoming drawbacks and favoring ease of handling, so as to maximize the potential benefits of EOs [84]. The use of cyclodextrins can lead to an increase in the solubility of poorly water-soluble drugs, improved bioavailability, increased stability due to the increased protection of the molecule included in the cyclodextrin and, finally, a decrease in volatility and an increase in half-life. Alternatively, nanoencapsulation can be considered. The latter improves biological activity by offering stability and protection against changing environmental conditions from outside [85]. The encapsulating matrix that can be used to effectively encapsulate essential oils consists of proteins, polysaccharides and lipids, which are derived from plant (starch, cellulose and gluten), animal (dextrin, chitosan and casein), marine, and microbial sources. These materials have the characteristics of being highly soluble in water, biodegradable and readily available [86,87]. Encapsulation methods, including ionic gelation, coacervation, liposomes, nanoemulsions and nanoprecipitation, could all be favorably employed. The above makes us realize that, although promising and remarkable, the efficacy results found for these EOs must be read as preliminary study data. Further laboratory and field studies are needed to arrive at a stable formulation that possesses, when administered in the field, a neutralization percentage of 90% or more, as recommended for substances of natural origin by the “Guideline on veterinary medicinal products controlling *Varroa destructor* parasitosis in bees” [88].

## 5. Conclusions

The results of the present article show that the hydrodistilled EOs from four selected Lamiaceae plants, which are native to the Calabria region in Italy, expressed interesting in vitro acaricidal activity. These EOs could be used singly or in combination to maintain *V. destructor* populations below the damage threshold. A residual toxicity test was used to ascertain the acaricidal activity of the EOs. Having given this test extremely positive results, one could speculate on a better ability of total exposure tests to neutralize mites. These types of tests would also be interesting to conduct. Field studies would also be needed to totally validate the efficacy of these EO species and to select the best vehicle for administering these products. Indeed, EOs evaporate rapidly or more slowly when exposed to exogenous environmental temperature factors. Field studies, outside controlled environmental laboratory conditions, should investigate the most suitable carrier for administration. Certainly, ours and other studies in line with it pave the way for a new concept of drug treatment in beekeeping that is more in line with an environmentally and consumer-friendly strategy.

## Figures and Tables

**Figure 1 vetsci-10-00701-f001:**
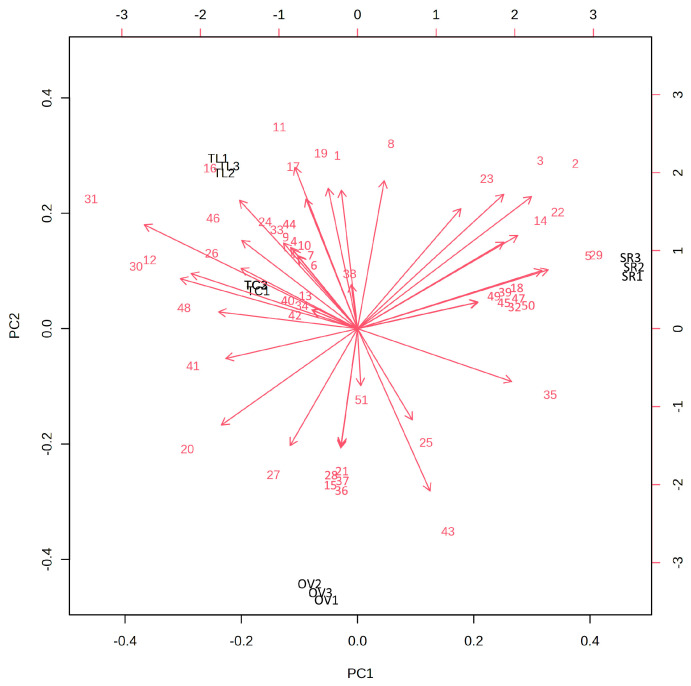
PCA biplot (PC1 vs. PC2) based on the chemical composition of the essential oils (EOs). OV, *Origanum vulgare* subsp. *viridulum* (Martrin-Donos) Nyman; SR, *Salvia rosmarinus* Schleid; TC, *Thymus capitatus* Hoffmanns. and Link; TL, *Thymus longicaulis* C. Presl. The serial numbers of phytochemicals are consistent with the peak numbering in Table 1.

**Figure 2 vetsci-10-00701-f002:**
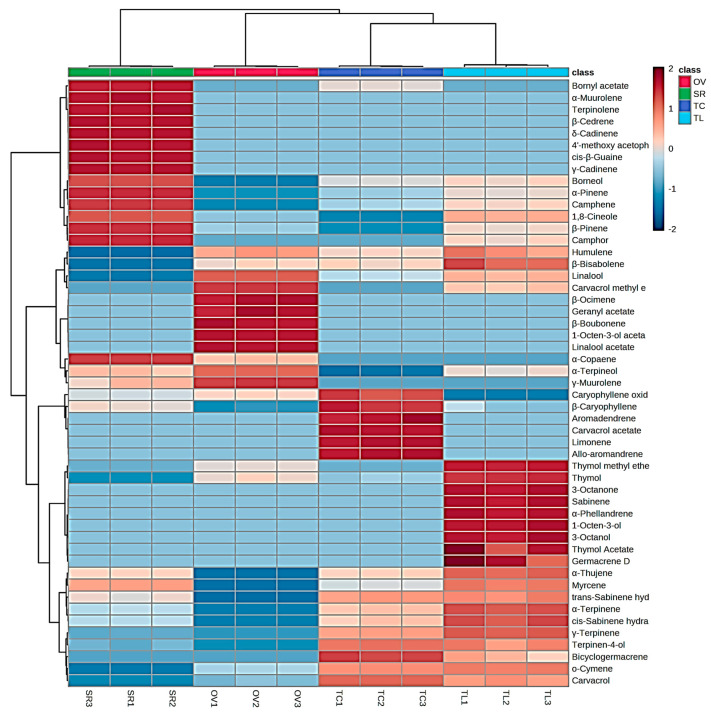
Heatmap of identified phytochemicals. Abbreviations are as follows: OV, *Origanum vulgare* subsp. *viridulum* (Martrin-Donos) Nyman; SR, *Salvia rosmarinus* Schleid; TC, *Thymus capitatus* Hoffmanns. and Link; TL, *Thymus longicaulis* C. Presl.

**Figure 3 vetsci-10-00701-f003:**
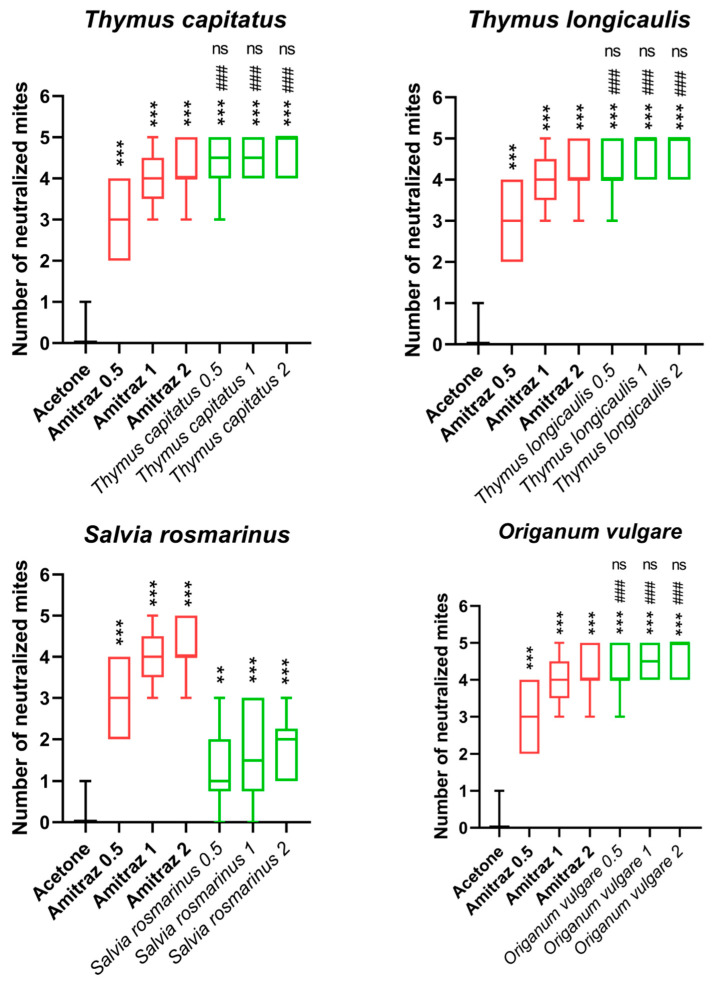
Effects of EO on neutralization of mites. *** *p* < 0.001 vs. acetone, ### *p* < 001 vs. Amitraz 0.5, ns (not significant) *p* > 0.05 vs. Amitraz 1 and Amitraz 2. Statistical analyses were performed using a one-way ANOVA and Bonferroni test for multiple comparisons. The symbols indicate the degree of statistical significance: three symbols indicate a value of *p* ≤ 0.001, two symbols *p* ≤ 0.01, “ns” when *p* > 0.05. The red color is used for the positive control blox pot, the green color for the essential oil box plot.

**Table 1 vetsci-10-00701-t001:** Chemical profile of investigated EOs.

No.	Compound ^1^	KI ^2^	KI ^3^	% ± SD				i.m. ^4^
				OV	SR	TC	TL	
1	α-Thujene	930	932	-	0.10 ± 0.00	0.10 ± 0.00	0.25 ± 0.01	GC-MS
2	α-Pinene	939	937	-	7.55 ± 0.03	0.09 ± 0.01	0.24 ± 0.01	GC, GC-MS
3	Camphene	954	955	-	2.54 ± 0.02	0.08 ± 0.01	0.21 ± 0.01	GC, GC-MS
4	Sabinene	975	970	-	-	-	0.20 ± 0.01	GC, GC-MS
5	β-Pinene	979	980	0.08 ± 0.00	6.48 ± 0.01	-	0.26 ± 0.01	GC, GC-MS
6	1-Octen-3-ol	980	981	-	-	-	0.49 ± 0.02	GC-MS
7	3-Octanone	986	984	-	-	-	2.22 ± 0.08	GC-MS
8	Myrcene	991	990	-	0.62 ± 0.02	0.27 ± 0.01	0.91 ± 0.06	GC, GC-MS
9	3-Octanol	993	995	-	-	-	0.74 ± 0.05	GC-MS
10	α-Phellandrene	1002	1001	-	-	-	0.35 ± 0.02	GC, GC-MS
11	α-Terpinene	1018	1015	-	0.10 ± 0.01	0.23 ± 0.02	0.95 ± 0.06	GC, GC-MS
12	o-Cymene	1022	1021	0.39 ± 0.01	-	2.16 ± 0.09	2.68 ± 0.13	GC, GC-MS
13	Limonene	1029	1028	-	-	0.60 ± 0.02	-	GC, GC-MS
14	1,8-Cineole	1031	1031	0.83 ± 0.02	46.92 ± 0.04	-	8.85 ± 0.50	GC, GC-MS
15	β-Ocimene	1050	1051	0.14 ± 0.01	-	-	-	GC-MS
16	γ-Terpinene	1062	1061	0.20 ± 0.00	0.28 ± 0.01	2.49 ± 0.09	5.86 ± 0.30	GC, GC-MS
17	*cis*-Sabinene hydrate	1068	1070	0.20 ± 0.00	0.56 ± 0.01	0.97 ± 0.08	2.55 ± 0.20	GC, GC-MS
18	Terpinolene	1088	1089	-	0.18 ± 0.01	-	-	GC-MS
19	trans-Sabinene hydrate	1097	1095	-	0.11 ± 0.01	0.22 ± 0.01	0.27 ± 0.03	GC-MS
20	Linalool	1098	1101	15.84 ± 0.00	0.28 ± 0.01	1.81 ± 0.12	5.14 ± 0.37	GC, GC-MS
21	1-Octen-3-ol acetate	1112	1119	0.38 ± 0.00	-	-	-	GC-MS
22	Camphor	1146	1142	-	8.49 ± 0.04	-	0.68 ± 0.08	GC, GC-MS
23	Borneol	1169	1164	-	11.96 ± 0.07	2.17 ± 0.05	2.75 ± 0.19	GC, GC-MS
24	Terpinen-4-ol	1177	1175	0.17 ± 0.00	0.23 ± 0.01	0.79 ± 0.02	0.70 ± 0.09	GC, GC-MS
25	α-Terpineol	1188	1188	5.13 ± 0.10	1.80 ± 0.18	0.15 ± 0.01	1.15 ± 0.07	GC, GC-MS
26	Thymol methyl ether	1235	1237	0.18 ± 0.01	-	-	5.10 ± 0.25	GC-MS
27	Carvacrol methyl ether	1244	1247	1.53 ± 0.02	-	-	0.30 ± 0.02	GC-MS
28	Linalool acetate	1257	1256	65.27 ± 0.72	-	-	-	GC, GC-MS
29	Bornyl acetate	1288	1289	-	2.74 ± 0.19	0.10 ± 0.00	-	GC, GC-MS
30	Thymol	1290	1271	0.76 ± 0.18	-	0.18 ± 0.03	31.67 ± 0.97	GC, GC-MS
31	Carvacrol	1299	1307	0.23 ± 0.06	-	54.74 ± 0.78	14.39 ± 0.68	GC, GC-MS
32	4′-methoxy acetophenone	1350	1349	-	0.09 ± 0.00	-	-	GC-MS
33	Thymol Acetate	1355	1357	-	-	-	1.63 ± 0.05	GS-MS
34	Carvacrol acetate	1372	1376	-	-	15.22 ± 0.38	-	GC-MS
35	α-Copaene	1376	1380	0.12 ± 0.01	0.48 ± 0.00	-	-	GC-MS
36	Geranyl acetate	1381	1385	0.25 ± 0.01	-	-	-	GC, GC-MS
37	β-Boubonene	1388	1388	0.15 ± 0.01	-	-	-	GC-MS
38	β-Caryophyllene	1408	1406	3.87 ± 0.09	6.46 ± 0.14	12.58 ± 0.59	5.18 ± 0.51	GC, GC-MS
39	β-Cedrene	1420	1417	-	0.11 ± 0.00	-	-	GC-MS
40	Aromadendrene	1441	1428	-	-	0.13 ± 0.01	-	GC-MS
41	α-Humulene	1454	1445	0.42 ± 0.04	-	0.25 ± 0.00	0.50 ± 0.11	GC-MS
42	Allo-aromandrene	1461	1453	-	-	0.10 ± 0.00	-	GC-MS
43	γ-Muurolene	1479	1476	2.36 ± 0.12	0.38 ± 0.02	-	-	GC-MS
44	Germacrene D	1480	1475	-	-	-	0.56 ± 0.02	GC-MS
45	cis-β-Guaine	1493	1491	-	0.15 ± 0.00	-	-	GC-MS
46	Bicyclogermacrene	1494	1492	-	-	1.32 ± 0.07	0.33 ± 0.01	GC-MS
47	α-Muurolene	1500	1499	-	0.09 ± 0.00	-	-	GC-MS
48	β-Bisabolene	1505	1511	0.99 ± 0.09	-	0.99 ± 0.06	2.90 ± 0.17	GC-MS
49	γ-Cadinene	1513	1510	-	0.28 ± 0.00	-	-	GC-MS
50	δ-Cadinene	1523	1521	-	0.68 ± 0.00	-	-	GC-MS
51	Caryophyllene oxide	1583	1569	0.50 ± 0.02	0.35 ± 0.00	2.27 ± 0.31	-	GC-MS

^1^ Components are reported according to their elution order on a polar column; ^2^ KI from the literature; ^3^ KI measured on a polar column; ^4^ identification method. OV, *Origanum vulgare* subsp. *viridulum* (Martrin-Donos) Nyman; SR, *Salvia rosmarinus* Schleid; TC, *Thymus capitatus* Hoffmanns. and Link; TL, *Thymus longicaulis* C.Presl.

**Table 2 vetsci-10-00701-t002:** Percentage and standard deviation (±) of *V. destractor* neutralization after treatments.

Concentration mg/mL	*O. vulgare viridulum* Mortality %	*T. capitatus* Mortality %	*T. longicaulis* Mortality %	*S. rosmarinus* Mortality %	Acetone Mortality %	Amitraz Mortality %
0.5 mg	86 (±13)	88 (±14)	84 (±13)	26 (±19)	2 S (±6)	60 (±14)
1 mg	90 (±11)	88 (±11)	94 (±10)	34 (±25)		67 (±3)
2 mg	94 (±10)	92 (±10)	94 (±10)	38 (±15)		93 (±10)

## Data Availability

Data are kept at Magna Græcia University of Catanzaro and are available upon request.

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
