# Peer review of "Chemical Profile of Essential Oils of Selected Lamiaceae Plants and In Vitro Activity for Varroosis Control in Honeybees *(Apis mellifera)"

_vetsci, 2023, doi:10.3390/vetsci10120701_

Round 1

Reviewer 1 Report

Comments and Suggestions for Authors

Dear authors,

Your article untitled "Pharmacological Activity and Phytochemical Profile of Essential Oils of Native Southern Italian Botanical Species Belonging to Lamiaceae Family for Sustainable Control of Varroosis in Honeybees" is an interesting in vitro study. However, some comments should be considered before publication.

Title

1.     The title is very long and thus it should be shortened. 

Introduction

1.     The introduction is very instructive and have a lot of information. However, there is too much information and it is very difficult to understand the important information. Could you summarize the main information?

2.     L126-132 these sentence should be introduce in the discussion section or earlier in the introduction

Author Response

Dear reviewer, thank you very much for taking the time to review our manuscript. We have responded, point by point, to all your questions and have integrated and corrected the text in the indicated parts. You will find the answers in bold.

Comments and Suggestions for Authors

Dear authors,

Your article untitled "Pharmacological Activity and Phytochemical Profile of Essential Oils of Native Southern Italian Botanical Species Belonging to Lamiaceae Family for Sustainable Control of Varroosis in Honeybees" is an interesting in vitro study. However, some comments should be considered before publication.

Title

  1. The title is very long and thus it should be shortened. 

R: Thank you for your advice. The title has been embraced as suggested.

Introduction

  1. The introduction is very instructive and have a lot of information. However, there is too much information and it is very difficult to understand the important information. Could you summarize the main information?

R: Thank you for your comment which helps us to improve the quality of the manuscript. The introduction has been shortened as suggested. In particular, sentences in the final part of the introduction have been eliminated.

  1. L126-132 these sentence should beintroduce in the discussion section or earlier in the introduction

R: Thank you very much for your comment. As suggested, the paragraph has been moved to the discussions.

Reviewer 2 Report

Comments and Suggestions for Authors

Dear authors,

Thank you for providing me the interesting scientific study  entitled vetsci-2733768 "Pharmacological Activity and Phytochemical Profile of Essential Oils of Native Southern Italian Botanical Species Belonging to Lamiaceae Family for Sustainable Control of Varroosis in Honeybees" by Bava et al submitted for publication to the Journal of Vet Scie

Firstly the title has to be changed, as it is long and somehow is misleading the readers

The phytocemical profile of the EOS, is just a GC-MS analysis of the volatiles from the plants. Mainly talking for phytochemical profile, we deal with full chemical analyses and even isolations and structural determinations

The aromatic plants used they are native of south Italy but also native to almost all mediterranean countries, so to my opinion can be kept in the text but not in the tile- proposed to be deleted

The pharmacological activity  tested, is in fact an in vitro test of the oils against Varroosis in honeybees, and  really still far to prove that could control this big problem in real conditions

-Proposed the title as" "chemical Profile of Essential Oils (or volatiles) of selected Lamiaceae plants  and iv vitro activities towards Control of Varroosis in Honeybees" - shorter and showing all content of the study

In Introduction part, 58 out of the total of 70 references are cited. Enormus number just for the Introduction (approx 72% of all refs)

Please confirm the Salvia rosmarinus Schleid. is more preferable than Salvia rosmarinus Spenn and maybe has to be cited, at least once in the text, that is syn with Rosmarinus officinalis

Lines  132-133 to be deleted, or better explained as all 4 essnial oils are very well studied and chemically analysed untill now (maybe not from Calabria but surely from many different neighbourhood geographic areas

In 2.1

Line 143 Labiatae to be changed in Lamiaceae 9as in the tile) Please confirm who has provided the botanical identification of the plant material

Line169-171 Please give more information aboiut the authentic reference compounds used ( origin ), the type of the library which was used

Line 195 Amitraz has been referred (also in the abstract), I could propose once in the text to explain the general use of amitraz, to give its full chemical name (N,N'-[(methylimino)dimethylidyne] di-2,4-xylidine, and some info on its solubility (not soluble in water and preferably diluted in aceton), used mainly as aerosol or impregnated stripes

For Table 1, I would propose: Chemical analyses of the studied essenial oils (EOs)

It was followed a full set comparison of the chemical contet of all 4 EOS, but not any comparison of these EOS with previous studied ones, as laready expressed before all 4 oils hve been studied many times before. Are the current analyses  comparable with previous ones? or what Also please provide refs to compare

In Line 268 please change "phytochemicals" with "metabolites"

In the Discussion please check Lines349-351 with previous 108-110, which are repeated. Keep up only once this information

In Lines 392-396 you analyse the potential interest of the use of cyclodextrins while you could also refer to the use of the modern method of  encapsulation of EOs in nanoparticles too (with appropriate refs)

In the Conclusion Part the lines 398-400 "The results of the present article show that EOs extracted from botanical species belonging to the Lamiaceae family, which are typical of the Calabria region, have good acaricidal efficacy." to be changed to " The results of the present article show that the EOs hydrodistilles from four selected Lamiaceae plants also native of the Calabria region in Italy, expresssed interesting  in vitro acaricidal activity"

Lines 409-411 please recontruct to show that the results are promissing but are needed further studied in the filed and in bee-hives, in order to talk for potential new drugs approach

After these corrections and amendments the manuscript could be accepted

Kind regards

Author Response

Dear reviewer, thank you for taking the time to review our manuscript. We have responded, point by point, to all your questions and have integrated and corrected the text in the indicated parts. You will find the answers in bold.

Dear authors,

Thank you for providing me the interesting scientific study  entitled vetsci-2733768 "Pharmacological Activity and Phytochemical Profile of Essential Oils of Native Southern Italian Botanical Species Belonging to Lamiaceae Family for Sustainable Control of Varroosis in Honeybees" by Bava et al submitted for publication to the Journal of Vet Scie

R: Thank you very much for your appreciation of our manuscript. Your suggested changes have been made to the text and you will find them highlighted in the revision mode and coloured in red

Firstly the title has to be changed, as it is long and somehow is misleading the readers

R: Thank you for your comment. The title has been changed according to your suggestion.

The phytocemical profile of the EOS, is just a GC-MS analysis of the volatiles from the plants. Mainly talking for phytochemical profile, we deal with full chemical analyses and even isolations and structural determinations

R: Thank you for your comment we have amended the text in agreement. Now you will find written chemical analysis and no longer phytochemical profile

The aromatic plants used they are native of south Italy but also native to almost all mediterranean countries, so to my opinion can be kept in the text but not in the tile- proposed to be deleted

R: We have modified the title according to your suggestion

The pharmacological activity  tested, is in fact an in vitro test of the oils against Varroosis in honeybees, and  really still far to prove that could control this big problem in real conditions

R: A sentence has been added in the discussions specifying that this is a preliminary study and further studies on the same oils are needed to validate its effectiveness. Thanks for this important comment

-Proposed the title as" "chemical Profile of Essential Oils (or volatiles) of selected Lamiaceae plants  and iv vitro activities towards Control of Varroosis in Honeybees" - shorter and showing all content of the study

R: Many thanks for your suggestion

In Introduction part, 58 out of the total of 70 references are cited. Enormus number just for the Introduction (approx 72% of all refs)

R: Thank you for your comment. Other references have been added in the text, in particular in the discussion section. There is now no more imbalance in the bibliographical references.

Please confirm the Salvia rosmarinus Schleid. is more preferable than Salvia rosmarinus Spenn and maybe has to be cited, at least once in the text, that is syn with Rosmarinus officinalis

R: The first choice, which is in the text, is the preferable one

Lines  132-133 to be deleted, or better explained as all 4 essnial oils are very well studied and chemically analysed untill now (maybe not from Calabria but surely from many different neighbourhood geographic areas

R: Thank you for your comment. The sentence has been deleted in accordance with the suggestion.

In 2.1

Line 143 Labiatae to be changed in Lamiaceae 9as in the tile) Please confirm who has provided the botanical identification of the plant material

R: The word was changed in accordance with the suggestion, and the change was confirmed by the members of the research team who were engaged in collecting and identifying plant species

Line169-171 Please give more information about the authentic reference compounds used ( origin ), the type of the library which was used

R: Thank you for your comment which helps us to improve the quality of the manuscript. The required information  has been added in the text.

Line 195 Amitraz has been referred (also in the abstract), I could propose once in the text to explain the general use of amitraz, to give its full chemical name (N,N'-[(methylimino)dimethylidyne] di-2,4-xylidine, and some info on its solubility (not soluble in water and preferably diluted in aceton), used mainly as aerosol or impregnated stripes

R: Thank you for your suggestion. A sentence about the solubility in organic solvents of essential oils and amitraz has been added in the materials and methods section of the manuscript

For Table 1, I would propose: Chemical analyses of the studied essenial oils (EOs)

R: Now amended accordingly

It was followed a full set comparison of the chemical contet of all 4 EOS, but not any comparison of these EOS with previous studied ones, as laready expressed before all 4 oils hve been studied many times before. Are the current analyses  comparable with previous ones? or what Also please provide refs to compare

R: We thank you for this comment which helps us to explain something we think is important. It is important to say that research results conducted with the same botanical species in different studies could lead to different results because the essential oils extracted from a botanical species have a different composition which is influenced by the characteristics of the soil in which the plant grows, by the exposure solar and the adaptations that the plant uses to better grow in the environment. The same comment was added as a sentence in the text.

In Line 268 please change "phytochemicals" with "metabolites"

R: Modified as suggested

In the Discussion please check Lines349-351 with previous 108-110, which are repeated. Keep up only once this information

R: Thanks for the advice, we corrected the discussions

In Lines 392-396 you analyse the potential interest of the use of cyclodextrins while you could also refer to the use of the modern method of  encapsulation of EOs in nanoparticles too (with appropriate refs)

R: Thank you for your suggestion that helps us improve the manuscript. Reference was made to nanoparticle encapsulation techniques as you suggested in the discussion section.

In the Conclusion Part the lines 398-400 "The results of the present article show that EOs extracted from botanical species belonging to the Lamiaceae family, which are typical of the Calabria region, have good acaricidal efficacy." to be changed to " The results of the present article show that the EOs hydrodistilles from four selected Lamiaceae plants also native of the Calabria region in Italy, expresssed interesting  in vitro acaricidal activity"

R: Modified as suggested

Lines 409-411 please recontruct to show that the results are promissing but are needed further studied in the filed and in bee-hives, in order to talk for potential new drugs approach

R: Many thanks for your comment. We have included a sentence in the discussions stating that this is a preliminary study and further laboratory and field studies are needed to confirm the efficacy of the tested essential oils.

Reviewer 3 Report

Comments and Suggestions for Authors

The topic of the article is very interesting. Finding new natural and eco-sustainable approaches against varroa would be a great goal for beekeeping. It would therefore have a practical implication. The experimental design is well organized and the statistical analysis is suitable for the results.

Minor comments:

-Abstract page 2 lines 49-52 If I'm not mistaken Salvia rosmarinus is not effective at high levels. Furthermore, in the next sentence the authors say that it has low levels of effectiveness. Please correct this sentence.

-Page 2 line 53 “Lamiaceae” instead of “Lamiacae”.

-Pages 2-3 lines 61-62, 73-74, 94-95, 96-97, 101-102, 103-105, bibliographical references are missing. Please add them.

-page 3 line 124 “in vitro” should be written in italics.

-page 3 lines 139-140 references to doctors who contributed to the work are generally not written in the materials and methods but as authors of the work or in the acknowledgments.

-page 4 lines 149, 159 in reference to the instruments please add model, brand and city.

-page 8  lines 298-304 this part describes the materials and methods for which it should be removed or moved to the materials and methods section.

-Figure 1 caption “vs” should be written in italics.

-page 10 lines 333-336, 351-354 , bibliographical references are missing. Please add them.

Comments on the Quality of English Language

I suggest reviewing English language.

Author Response

Dear reviewer, thank you very much for taking the time to review our manuscript. We have responded, point by point, to all your questions and have integrated and corrected the text in the indicated parts. You will find the answers in bold.

The topic of the article is very interesting. Finding new natural and eco-sustainable approaches against varroa would be a great goal for beekeeping. It would therefore have a practical implication. The experimental design is well organized and the statistical analysis is suitable for the results.

Minor comments:

-Abstract page 2 lines 49-52 If I'm not mistaken Salvia rosmarinus is not effective at high levels. Furthermore, in the next sentence the authors say that it has low levels of effectiveness. Please correct this sentence.

R: Thank you very much for pointing out this important oversight. We have corrected it in accordance with your suggestion.

-Page 2 line 53 “Lamiaceae” instead of “Lamiacae”.

R: Now amended

-Pages 2-3 lines 61-62, 73-74, 94-95, 96-97, 101-102, 103-105, bibliographical references are missing. Please add them.

R: The bibliographical references have been added

-page 3 line 124 “in vitro” should be written in italics.

R: Now amended

-page 3 lines 139-140 references to doctors who contributed to the work are generally not written in the materials and methods but as authors of the work or in the acknowledgments.

R: The sentence has been deleted as suggested

-page 4 lines 149, 159 in reference to the instruments please add model, brand and city.

R: The required information has been added

-page 8  lines 298-304 this part describes the materials and methods for which it should be removed or moved to the materials and methods section.

R: The sentence has been removed as suggested

-Figure 1 caption “vs” should be written in italics.

R: Now amended

-page 10 lines 333-336, 351-354 , bibliographical references are missing. Please add them.

R: The bibliographical references have been added as suggested.
